# Inhibition of Aflatoxin B1 Synthesis in *Aspergillus flavus* by Mate (*Ilex paraguariensis*), Rosemary (*Rosmarinus officinalis*) and Green Tea (*Camellia sinensis*) Extracts: Relation with Extract Antioxidant Capacity and Fungal Oxidative Stress Response Modulation

**DOI:** 10.3390/molecules27238550

**Published:** 2022-12-05

**Authors:** Anthony Al Khoury, André El Khoury, Ophélie Rocher, Pamela Hindieh, Olivier Puel, Richard G. Maroun, Ali Atoui, Jean-Denis Bailly

**Affiliations:** 1Centre d’Analyse et de Recherche, Unité de Recherche Technologies et Valorisations Agro-Alimentaires, Faculté des Sciences, Campus des Sciences et Technologies, Université Saint-Joseph de Beyrouth, Mar Roukos, Matn 1104-2020, Lebanon; 2Toxalim (Research Center in Food Toxicology), Université de Toulouse, INRAE, ENVT, EI-Purpan, 31300 Toulouse, France; 3Laboratory of Microbiology, Department of Life and Earth Sciences, Faculty of Sciences, Hadath Campus, Lebanese University, P.O. Box 5, Beirut 1104, Lebanon; 4Laboratoire de Chimie Agro-Industrielle (LCA), Université de Toulouse, INRA, INPT, ENVT, 4 Allée Emile Monso, 31030 Toulouse, France

**Keywords:** aflatoxin B1, inhibition, mate, green tea, phenolic compounds, catalase, superoxide dismutase

## Abstract

Plant extracts may represent an ecofriendly alternative to chemical fungicides to limit aflatoxin B1 (AFB1) contamination of foods and feeds. Mate (*Ilex paraguariensis*), rosemary (*Romarinus officinalis*) and green tea (*Camellia sinensis*) are well known for their beneficial properties, which are mainly related to their richness in bioactive phenolic compounds. AFB1 production is inhibited, with varying efficiency, by acetone/water extracts from these three plants. At 0.45 µg dry matter (DM)/mL of culture medium, mate and green tea extracts were able to completely inhibit AFB1 production in *Aspergillus flavus,* and rosemary extract completely blocked AFB1 biosynthesis at 3.6 µg DM/mL of culture medium. The anti-AFB1 capacity of the extracts correlated strongly with their phenolic content, but, surprisingly, no such correlation was evident with their antioxidative ability, which is consistent with the ineffectiveness of these extracts against fungal catalase activity. Anti-AFB1 activity correlated more strongly with the radical scavenging capacity of the extracts. This is consistent with the modulation of SOD induced by mate and green tea in *Aspergillus flavus*. Finally, rutin, a phenolic compound present in the three plants tested in this work, was shown to inhibit AFB1 synthesis and may be responsible for the anti-mycotoxin effect reported herein.

## 1. Introduction

Aflatoxin B1 (AFB1) is a carcinogenic, mutagenic, teratogenic and immunosuppressive mycotoxin [1] produced by several species of the genus *Aspergillus* that belong primarily to the *Flavi* section [2]. An estimated 4.5 billion people are exposed to aflatoxin contamination from various foodstuffs, especially in developing countries where the climate may favor AFB1 synthesis and regulations may be weak [3,4]. Moreover, ongoing climate change has led to the emergence of aflatoxin contamination in previously unaffected areas [5,6,7]. If environmental conditions allow the development of aflatoxigenic strains, aflatoxin production can occur at various stages during crop production, namely, pre-harvest, transportation and post-harvest during storage [8,9].

AFB1 contamination is commonly prevented by using pesticides to limit fungal development in the field and by correctly drying grains to avoid contamination during storage [10,11]. However, the efficacy of these methods may be limited due to the difficulty of controlling key environmental factors [12]. Moreover, the use of chemical pesticides is suspected to be harmful to human health, the environment, and biodiversity [13,14,15].

Thus, significant research now focuses on identifying alternative methods to reduce the production of AFB1 while preserving ecological diversity and human health. These approaches are based on two main strategies, the first of which focuses on biological control using atoxigenic fungal strains, bacteria, or yeasts as natural competitors with the toxigenic fungi [16,17,18,19]. However, the possible long-term effects and sustainability of adding or enhancing a specific single microorganism over a complex microbiota remains unclear [20].

The second strategy uses plant-based extracts or natural compounds to inhibit aflatoxin production by blocking fungal development or specifically limiting toxin biosynthesis [21]. In fact, several plants have been found to inhibit AFB1 production with limited or no impact on fungal growth, which could be a way to improve food safety without impacting biodiversity [21,22,23]. The antioxidative potential of many of these extracts or compounds was found to be related to their impact on AFB1 synthesis [24,25]. Indeed, some genes involved in the fungal response to oxidative stress are also linked to the regulation of the AFB1 gene cluster via the modulation of its two internal regulators (*aflR* and *aflS*) [26]. In some works, antioxidant molecules were shown to alter the intracellular balance between reactive oxygen species and free radical scavengers, and oxidative stress-inducing factors were shown to promote AFB1 biosynthesis [26].

Mate (*Ilex paraguariensis*), rosemary (*Rosmarinus officinalis)* as well as green tea (*Camellia sinensis*) are recognized for their high content of bioactive compounds, especially phenolic compounds. Yerba mate (or simply “mate”) is native to South America [27]. Mate tea is made through the infusion of leaves, and it is recognized worldwide for its stimulant and neuroprotective effects as well as its richness in bioactive compounds such as phenolic acids, which give it its antioxidant properties [28]. Rosemary is an evergreen shrub of Mediterranean origin [29] and is popularly used in cooking and for infusion. It is also known to be rich in antioxidant molecules and may be used to support the immune system [29]. The green tea tree is native to South East Asia [30], and the infusion of green tea leaves is the most consumed drink in the world after water [31]. Green tea extracts display a significant amount of catechin, which is a powerful antiradical of the flavanol family [32].

The purpose of this study was to evaluate the anti-aflatoxigenic effects of these plants’ extracts on *Aspergillus flavus* and to determine whether such effects correlate with the phenolic content of the extracts, their antioxidant properties, and/or their impact on enzymatic activities linked to the fungal oxidative stress responses.

The results show that all three extracts inhibited AFB1 production with varying efficiencies. Although the anti-AFB1 capacity of the extracts correlated strongly with their phenolic content, no such correlation was observed with their antioxidative ability, which is consistent with the extracts’ inability to reduce fungal catalase activity. However, anti-AFB1 activity correlated more strongly with the radical scavenging capacity of the extracts and the modulation of superoxide dismutase activity (SOD) induced by mate and green tea in *Aspergillus flavus*.

## 2. Results and Discussion

### 2.1. Choice of Extraction Solvent

Mate was used to obtain the optimal solvent mixture for extracting phenolic compounds from plants. Figure 1 demonstrates that acetone/water (50:50 *v:v*) is the best solvent mixture for extracting polyphenols from mate, so this solvent mixture was also used to prepare the other plant extracts. This is consistent with the results of Van Ngo et al. [33], who tested various solvents to determine their capacity to extract bioactive compounds from *Salacia chinensis* root.

### 2.2. Phenolic Content and Antioxidative Activity of Extracts

#### 2.2.1. Total Phenolic Content

Figure 2 shows the total polyphenol content of the three plant extracts, as determined by the Folin–Ciocalteu method.

Green tea extract contains the most polyphenol, with 10.25 mg gallic acid equivalents (GAE)/mL of extract, followed by mate and rosemary with 5.29 mg and 2.5 GAE/mL of extract, respectively. These values correspond to 102.1, 53.5, and 24.3 mg GAE/g dry matter (DM), respectively. These results are consistent with those reported for mate and green tea [34,35], given that phenolic contents depend strongly on the species of plant and the extraction procedure [36].

Compared with other plants or extracts, all three plants used here are rich in phenolic compounds. For instance, *Baphia nitida* (camwood) or *Phoenix dactylifera* fruits (dates), which are usually considered rich in phenolic compounds, contain 7.43 and 5.66 mg GAE/g DM, respectively [37]. *Vitis vinifera* (grape) fruit seeds, also considered as important sources of polyphenols, contain 7.5–40.4 mg GAE/g DM [38].

#### 2.2.2. Total Antioxidant Activity of Plant Extracts

Phosphomolybdenum reduction assays were used to evaluate the global antioxidant capacity of the plant extracts; the results are recapitulated in Figure 3. At a concentration of 0.0281 mg DM/mL of extract, the antioxidant capacity of the three extracts was 936, 867 and 1091 μg ascorbic acid equivalent for mate, rosemary, and green tea, respectively. Increasing the extract concentration further increased the antioxidant capacity of green tea above that of mate and rosemary. However, for the two highest concentrations tested (3.6 and 7.2 mg DM/mL of extract), mate became the strongest antioxidant of the three extracts tested, with 10.9 mg ascorbic acid equivalent.

These values are difficult to compare with previously reported values due to the large variation in sample preparation and how the results are expressed. However, these values are quite high if we compare them with available data on *Piper betle* (betel), which has an antioxidative capacity of 0.115 mg ascorbic acid equivalent/mL of plant extract [39]. Another study conducted on *Lysimachia foenum-graecum* extract reported an antioxidative activity of 40.16 µg ascorbic acid equivalent/mL plant extract [40].

#### 2.2.3. Antiradical Activity of Plant Extracts

The antiradical activity corresponded to the capacity of the extract components to neutralize reactive oxygen species, and, more specifically, superoxide anions or hydroxyl radicals [41]. As shown in Figure 4, at the highest concentration tested, mate and green tea extracts had a similar radical-scavenging ability, reducing more than 90% of the DDPH. The reaction was saturated at a concentration of 1.8 mg DM/mL of extract. In contrast, the antiradical activity of rosemary was lower, with a reduction of 75% of the DDPH radical at the highest concentration tested. At lower concentrations, rosemary was always the least active extract of the three plants under study, whereas green tea had a significantly higher antiradical activity than mate. As an illustration, at a concentration of 0.9 mg DM/mL of extract, rosemary extract reduced about 13% of DDPH, whereas mate and green tea reduced 66% and 83% of DDPH, respectively. The IC_50_ (i.e., the concentration at which 50% of DDPH is reduced) was 381, 276, and 5076 µg DM/mL of extract for mate, green tea, and rosemary, respectively. These values are consistent with those previously published for both mate and green tea [35,42].

Comparing these results with those obtained from other plants shows that the radical-scavenging abilities of mate and green tea extracts are relatively significant [43]. For instance, IC_50_ values of 62.6, 126, and 271.5 µg DM/mL of extract were reported for Kauri, Ghanagete and Bagerhati, respectively, which are three varieties of *Piper betle* [39]. *Cuminum cyminum* and *Coriandrum sativum* essential oils have IC_50_ = 494 and 756 µg/mL, respectively [44].

### 2.3. Impact of Plant Extracts on the Growth of Aspergillus flavus and on the Production of AFB1

Table 1 shows how increasing the concentration of plant extract affects fungal development.

Mate extract produced only a minor effect on the fungal radial growth, even at the highest concentration tested. Both rosemary and green tea extracts more strongly affected fungal development at high concentrations, reducing fungal radial growth by 16% and 18%, respectively, at 7.2 mg DM/mL of culture medium.

Although all extracts induced a dose-dependent reduction in AFB1 synthesis, major differences remained (Figure 5). For mate and green tea, AFB1 was no longer detectable in the culture medium at concentrations exceeding 0.45 mg DM/mL of culture medium. For rosemary, the anti-AFB1 effect was lower, and complete inhibition occurred at concentrations greater than 3.6 mg DM/mL of culture medium. To better evaluate how mate and green tea affect AFB1, lower concentrations were tested (Figure 6). The results show that mate extract more efficiently inhibited AFB1: at 0.0625 mg DM/mL of culture medium, AFB1 production was 52% and 89% that of the untreated control sample for mate and green tea, respectively. IC_50_ was calculated to be 0.067, 0.639 and 0.115 mg DM/mL of culture medium for mate, rosemary, and green tea extracts, respectively. Compared with other extracts, mate extract had high anti-AFB1 potential. For example, IC_50_ values recorded for *Satureja khozistanica* and *Satureja macrosiphonia* were approximately 0.06 and 0.45 mg DM/mL of culture medium, respectively [45]. In addition, Shukla et al. [46] reported that, for AFB1 inhibition, the IC_50_ for *Callistemon lanceolatus* was approximately 0.273 mg DM/mL of culture medium. Similarly, the IC_50_ for *Cuminum cyminum* (cumin), *Coriandrim sativum* (coriander), *Micromaeria graeca* (hyssop), and *Mimosa tenuiflora* (mimosa) ranged from 0.3 to 0.625 mg DM/mL of culture medium [22,23,44]. For these plant extracts, total inhibition of AFB1 production required concentrations as high as 1.25, 1.5, and 15 mg DM/mL of culture medium for *Cuminum cyminum* and *Coriander sativum* essential oils and *Micromeria graeca* aqueous extract, respectively [22,44]. Note that the IC_50_ of mate extract approaches that reported for *Cannabis sativa* flower extract (0.056 mg DM/mL of culture medium) [47].

### 2.4. Effect of Plant Extracts on Oxidative Stress Enzymatic Response in Aspergillus flavus

To evaluate how our three tested plant extracts affect the oxidative stress enzymatic response in *Aspergillus flavus*, the catalase and super oxide dismutase activities of the fungus were measured after exposure to two concentrations of the three plant extracts. The first concentration tested was 0.225 mg DM/mL of culture medium, and the second corresponded to the IC80 for each extract.

Figure 7 shows that exposure of *A. flavus* to the plant extracts at these concentrations did not significantly modify the catalase activity, regardless of the extract considered.

In contrast, incubation of *A. flavus* with 0.225 mg DM/mL of culture medium of the three extracts tended to reduce the SOD enzymatic activity (Figure 8A). This inhibition became significant when exposing *A. flavus* to green tea extract concentrations corresponding to IC_80_. However, at the mate extract concentration inhibiting the production of AFB1 by 80%, the SOD activity of *A. flavus* increased significantly (Figure 8B).

Such differential responses of oxidative-stress enzymes to antioxidant compounds have been reported for piperine, a well-known AFB1 inhibitor that increased catalase activity but had no significant effect on SOD [25]. In contrast, β-glucans of *Lentinula edodes* (Shiitake) inhibited AFB1 production and increased SOD enzymatic activity [48], as observed here for mate.

### 2.5. Correlation between Anti-AFB1 Potential of the Three Extracts and Phenolic Content, Antioxidant and Antiradical Activities

We used a Pearson correlation analysis to determine whether the phenolic content, antioxidant potential and antiradical activity of the three extracts correlated with their anti-aflatoxigenic effects. The resulting correlation coefficients are presented in Table 2.

The correlation analysis showed that the anti-aflatoxigenic activity of the plant extracts correlated strongly with total phenolic content. Antiradical activity also correlated strongly with anti-aflatoxigenic potential for the two most active extracts, mate and green tea. In contrast, anti-aflatoxin ability correlated only weakly with the global antioxidant activity of the extracts. These results are consistent with results obtained for enzymatic activities. SOD, which is the first line of defense against free radicals, was impacted by the extracts, whereas catalase, which is more of a defensive mechanism against peroxides, was not impacted by the extracts. 

### 2.6. Impact of Rutin on AFB1 Synthesis in Aspergillus flavus

Some of the phenolic compounds that are present in the three plants under study have already been identified as AFB1 inhibitors when used as pure compounds in culture. This was the case for caffeic acid, quercetin, and, to a lesser extent, chlorogenic acid. These phenolic compounds are present in rosemary [49,50], green tea [51], and mate [52] and are potential inhibitors of AFB1 synthesis [53,54,55]. However, considering the amounts of phenolic compounds present in the plants, the concentrations required for these molecules to inhibit AFB1 synthesis are much greater than those present in our extracts. However, another phenolic compound, rutin, appeared in all three plants under study [50,51,56]. Since this molecule has strong antiradical activity [57], we tested its ability to inhibit AFB1 in *A. flavus*.

The addition of 0.1 µg/mL of culture medium of pure rutin led, after 7 days of incubation at 27 °C, to an 88% inhibition of AFB1 production compared with an untreated control culture of *A. flavus* NRRL 62477, with a very limited impact on fungal growth (2% inhibition compared to control). Increasing the concentration of rutin to 0.5 µg/mL of culture medium only increased AFB1 inhibition by 2% to reach 90% inhibition, although growth was more impacted since the colony diameter of exposed cultures was 10% less than the diameter of control cultures. Since this phenolic compound is much more present in mate and green tea (about 5–10 and 1 mg/g DM, respectively) [34,36] than in rosemary (about 0.3 mg/g DM) [58], it may participate in the strong AFB1 inhibition by these two plant extracts. Further investigations are now underway to test this hypothesis.

## 3. Materials and Methods

### 3.1. Chemicals

All solvents were HPLC grade and purchased from Thermo Fisher Scientific (Illkirch, France).

### 3.2. Preparation of Plant Extracts

Plant material was purchased in a local market in Beirut (Lebanon). Dried leaves of the three plants were used to prepare the extracts.

To determine the solvent mixture required to optimize the extraction of phenolic compounds from the plants, 9 g of dried ground mate were extracted for 2 h in a water bath at 50 °C with 90 mL of one of the following solvents: distilled water, acetone–water (25:75 *v*:*v*), acetone–water (50:50 *v*:*v*), acetone–water (85:15 *v*:*v*), ethanol–water (50:50 *v*:*v*), methanol–water (50:50 *v*:*v*), or methanol–water (75:25 *v*:*v*). Extracts were then filtered through a sterile gauze, followed by a second filtration through a Whatman 1PS phase separator (GE Healthcare Life Sciences, Vélizy-Villacoublay, France). Organic solvent was evaporated by using a rotary evaporator (Büchi, Flawil, Switzerland) at 60 °C, then the solvent volume was increased to 90 mL by adding distilled water. Finally, the extracts were filtered again through 22 µm nylon syringe filters (Sigma-Aldrich, Darmstadt, Germany) before being stored at 4 °C until use.

For further experiments, 25 g each of mate, rosemary, and green tea leaves were ground with an electrical grinder. Nine grams of each powder were added to a 150 mL Erlenmeyer flask containing 90 mL of acetone–water (1:1, *v*:*v*) mixture, and extraction was performed as describe above.

### 3.3. Determination of Total Phenolic Content

The total phenolic content of the extracts was determined by applying the Folin–Ciocalteu technique as per Singleton et al. [59] and Ainsworth and Gillespie [60]. A volume of 0.8 mL of sodium carbonate (4.25% *w*/*v*) was added to 0.2 mL of each plant extract and mixed with 1 mL of the previously diluted 1/10 Folin–Ciocalteu reagent (Sigma-Aldrich, Darmstadt, Germany). Samples were incubated for 2 h, following which their absorbances at 750 nm were determined by using a UV-Vis spectrophotometer (Biochrom Ltd., Cambridge, UK). The total phenolic content was expressed in mg GAE/g DM based on a gallic acid calibration curve.

### 3.4. Determination of Total Antioxidant Activity

The antioxidant activity of the extracts was measured by using the phosphomolybdenum reduction assay as per Abi-Khattar et al. [61]. The extract concentrations tested were the same as those used in the *A. flavus* cultures. One hundred microliters of extract at each concentration were added to 1 mL of a reagent solution consisting of 28 mM of sodium phosphate, 4 mM of ammonium molybdate, and 0.6 M of sulfuric acid. After incubating for 90 min in a water bath at 95 °C, the absorbance was measured at 685 nm. An ascorbic acid calibration curve was used to express the antioxidant activity of the extracts in μg of ascorbic acid equivalent per milliliter.

### 3.5. Radical Scavenging Ability of Plant Extracts

The radical-scavenging capacity of the extracts was evaluated by using 2,2-diphenyl-1-picrylhydrazyl (DPPH) reduction [61]. Fifty microliters of extracts at various concentrations were added to 1.45 mL of DPPH (0.06 mM) (Sigma-Aldrich, St-Quentin Fallavier, France). The mixtures were held in the dark at room temperature for 30 min, following which the absorbance was measured at 515 nm. The radical-scavenging ability (RSA) in percentage was calculated as follows:RSA% = [(absorbance of control − absorbance of sample)/absorbance of control] × 100

### 3.6. Impact of Extracts on Fungal Growth and AFB1 Synthesis

The extracts were tested on *Aspergillus flavus* NRRL 62477 [62] to determine their effect on fungal growth and aflatoxin synthesis. A calibrated spore suspension in a 0.05% tween 80 solution (10^5^ spores/mL) was prepared from a 7 day culture on a malt extract agar (MEA) medium (Biokar Diagnostics, Allone, France) at 27 °C. One thousand spores were inoculated centrally on MEA media treated with the following volumes of extract, (corresponding to increasing dry matter (DM) content per mL of culture medium): 50 µL (0.225 mg DM/mL of culture medium); 100 µL (0.45 mg DM/mL of culture medium); 200 µL (0.9 mg DM/mL of culture medium); 400 µL (1.8 mg DM/mL of culture medium); 800 µL (3.6 mg DM/mL of culture medium); and 1600 µL (7.2 mg DM/mL of culture medium). For mate and green tea, smaller concentrations were also tested: 6.25 µL (0.028 mg DM/mL of culture medium), 12.5 µL (0.056 mg DM/mL of culture medium) and 25 µL (0.115 mg DM/mL of culture medium). Controls were prepared by adding the same volumes of sterile distilled water instead of extracts. The Petri dishes were then incubated at 27 °C in the dark for 7 days. The colony diameter was measured at the end of the experiment to evaluate the impact of the extracts on fungal growth.

### 3.7. AFB1 Quantification

After 7 days of incubation, the culture medium was cut into small samples and mixed with 30 mL of chloroform prior to agitation for 2 h on a horizontal shaker at 150 rpm at room temperature. The samples were filtered through a Whatman 1PS phase separator (GE Healthcare Life Sciences, Vélizy-Villacoublay, France). Two mL of the filtrate were evaporated at 50 °C until dry and suspended in 2 mL methanol. Each sample was filtered again through a 0.45 µm syringe filter before HPLC analysis (Thermo Scientific Fisher, Villebon-Sur-Yvette, France). AFB1 production was quantified via a reverse-phase HPLC (Waters Alliance, Milford, MA, USA) coupled with a fluorescent detector. Separation was achieved by using a C18 column 5 μm, 250 × 4.6 mm (Supleco, Bellefonte, PA, USA) fitted with an HS C18, Supelguard Discovery, 20 × 4 mm, 5 μm, precolumn (Supelco, Bellefonte, PA, USA). The column compartment temperature was held at a constant 40 °C. The mobile phase consisted of water-methanol-nitric acid 4M (55:45:0.35 *v*/*v*/*v*) to which potassium bromide (KBr) at a concentration of 119 mL/L was added extemporaneously [63]. The flow rate was 0.8 mL/min, the injection volume was 20 μL, and the run time was 35 min for each sample. The excitation and emission wavelengths were 360 and 430 nm, respectively. The retention time for AFB1 was 16 min. A standard calibration curve was used to quantify AFB1. The limit of detection (LOD) and the limit of quantification (LOQ) of the method were 3.4 ng/L and 15 ng/L, respectively.

### 3.8. Analysis of Fungal Enzymatic Activities

#### 3.8.1. Sample Preparation

Two concentrations of each extract were used: 50 µL (0.225 mg DM/mL of culture medium) and the concentration causing 80% inhibition of AFB1 production (IC_80_). The IC_80_ values for mate, rosemary, and green tea were 29.4 µL (0.16 mg DM/mL of culture medium), 200.8 µL (1.11 mg DM/mL of culture medium) and 50.9 µL (0.28 mg DM/mL of culture medium), respectively. Controls were prepared by adding similar volumes of sterile distilled water instead of extracts to 20 mL of MEA. Prior to inoculation, culture media containing plant extracts were covered with a sterile cellophane layer to allow the removal of mycelium for enzymatic analysis (Hutchinson, Chalette-sur-Loing, France), as previously described [25]. A calibrated spore suspension (10^3^ spores) was then centrally inoculated on the media and incubated at 27 °C in the dark for 4 days.

#### 3.8.2. Measurement of Superoxide Dismutase Activity

Two hundred mg of mycelium were collected on plates using sterile scalpels and suspended in 1 mL of 20 mM HEPES buffer (pH 7.2; 1 mM EGTA, 210 mM mannitol, and 70 mM sucrose). The samples were frozen in liquid nitrogen to preserve enzymes, transferred to lysing matrix D tubes (1.4 mm ceramic spheres, MPBio, Eschwege, Germany), and then homogenized using Precellys^®^24 (Bertin Technologies, Montigny-le-Bretonneux, France) for two cycles of 10 s each. The samples were then centrifuged for 15 min at 10,000 rpm at 4 °C to prevent degradation. Supernatants containing proteins and enzymes were collected. Proteins were quantified by using a Pierce BCA Protein Assay Kit (Thermo Scientific, Villebon-Sur-Yvette, France), and superoxide dismutase activity was determined by using a commercial kit (Superoxide Dismutase (SOD-706002), Interchim, Montluçon, France) following the manufacturer’s instructions. Sample absorbance was determined at 450 nm by using an ELISA plate reader (Spectra thermo scan, Tecan, NC, USA). The total protein content was used to normalize enzymatic activity, and the results were expressed in U mL^−1^ per milligram of protein, where U (Units) refers to the amount of enzyme required to dismutate 50% of the superoxide radical.

#### 3.8.3. Measurement of Catalase Activity

Mycelium was collected and prepared as for the SOD assays but suspended in 1 mL of a 50 mM potassium-phosphate buffer (pH 7.0 containing 1 mM EDTA). After centrifugation for 15 min at 10,000 rpm and 4 °C, the supernatants were collected. Proteins were measured by using a Pierce BCA Protein Assay Kit (Thermo Scientific, Villebon-Sur-Yvette, France), and catalase activity was determined by using a commercial kit (Catalase (CAT-707002), Interchim, Montluçon, France) following the manufacturer’s instructions. Sample absorbance was determined at 540 nm by using an ELISA plate reader (Spectra thermo scan, Tecan, NC, USA). Total protein content was used to normalize activity, and results were expressed in nmol min^−1^ mL^−1^ mg^−1^ of protein.

### 3.9. Statistical Analysis

Results are expressed as mean values of three (or six) distinct experiments +/− SD. STATGRAPHICS^®^ Centurion XV (Statgraphics 18, The Plains, VA, USA) was used for the analysis of variance, least significant difference tests and Pearson correlation analysis. The IC_50_ (IC_80_) values for the different extracts corresponded to concentrations inhibiting AFB1 production by 50% (80%) compared with untreated control cultures. IC_50_ and IC_80_ were calculated by using the Compusyn software (Compusyn 229 inc, Paramus, NJ, USA; https://combosyn.com, accessed on 28 November 2022).

## 4. Conclusions

This study demonstrated that mate, rosemary, and green tea extracts inhibit AFB1 production in *Aspergillus flavus*. Their anti-aflatoxigenic properties correlated with the extract content of polyphenols and, to a lesser extent, with their radical-scavenging ability but not with their global antioxidant capacity. These results were confirmed by the extracts not affecting fungal catalase activity, whereas SOD activity was modified by mate and green tea extracts. Rutin, present in all three plants, is a phenolic compound with strong anti-radical activity and may contribute to the anti-aflatoxigenic properties of these extracts.

## Figures and Tables

**Figure 1 molecules-27-08550-f001:**
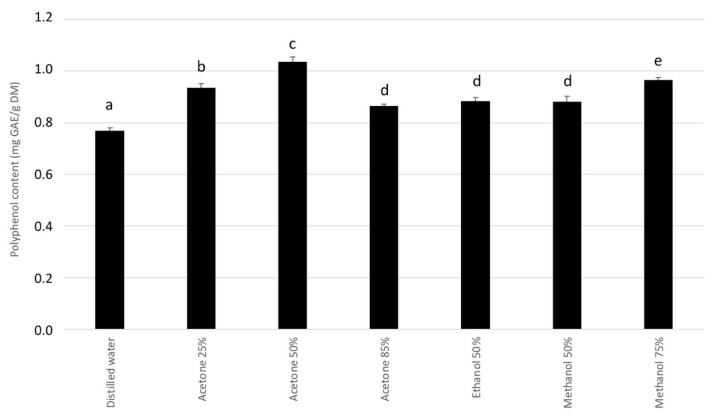
Total polyphenol content from mate as a function of the solvent mixture used. The total phenolic content of mate extracts was determined by using the Folin–Ciocalteu method. The results are expressed in gallic acid equivalent (GAE) and given as the mean +/− standard deviation (SD) of three experiments. The different letters labeling the histogram bars indicate significant differences between the solvents used for extraction (*p*-value < 0.05).

**Figure 2 molecules-27-08550-f002:**
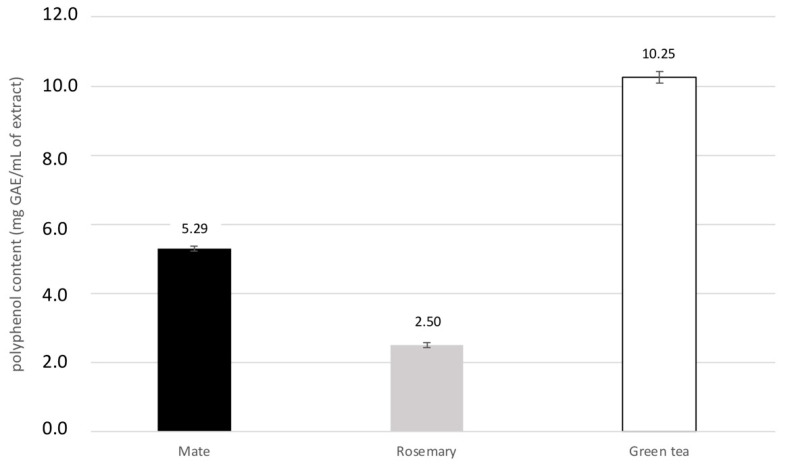
Polyphenol content of acetone–water (1:1) extract from mate (black bar), rosemary (grey bar) and green tea (white bar). Mean +/− SD of six replicates is shown, and the results are expressed in mg GAE/mL of extract.

**Figure 3 molecules-27-08550-f003:**
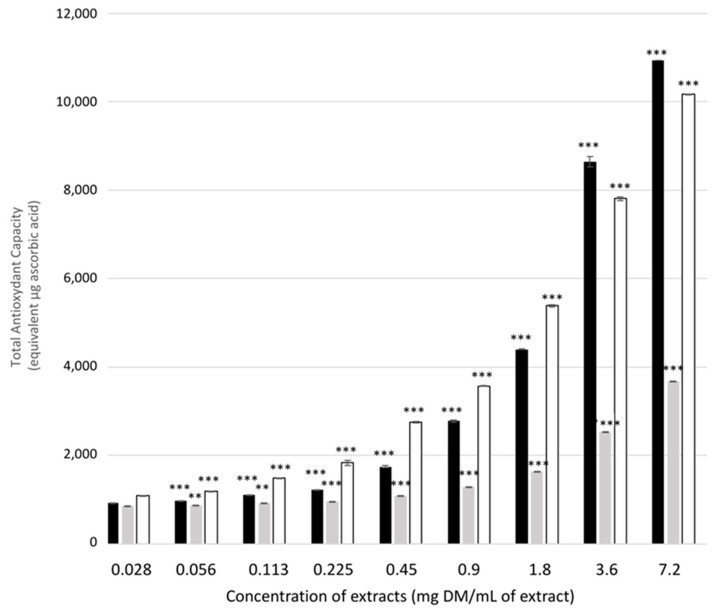
Total antioxidant capacity of the three plant extracts measured by phosphomolybdenum reduction and expressed in μg ascorbic acid equivalents. Mate: black bars, rosemary: grey bars, green tea: white bars. Shown are the means of six replicates +/− SD. Significant differences between the different concentrations of the same plant extract are indicated by ** for *p*-value < 0.01 and by *** for *p*-value < 0.001.

**Figure 4 molecules-27-08550-f004:**
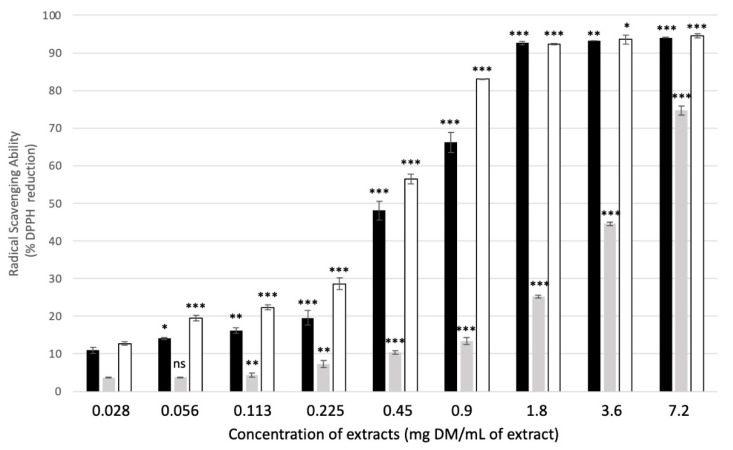
Radical scavenging ability (RSA) of the three plant extracts under study expressed in percentage of DPPH reduced: mate: black bars, rosemary: grey bars, green tea: white bars. Shown are the means of six replicates +/− SD. Significant differences between different concentrations of the same plant extract are indicated by * for *p*-value< 0.05, ** for *p*-value < 0.01, and *** for *p*-value < 0.001. ns: not significant.

**Figure 5 molecules-27-08550-f005:**
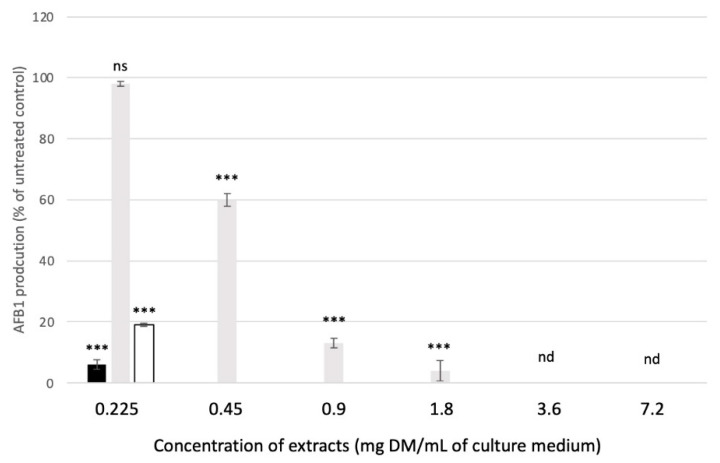
AFB1 production when exposed to various concentrations of mate (black bars), rosemary (grey bars) and green tea (white bars) extracts. ns: not significant. nd: not detected for the three extracts. Results correspond to the mean +/− SD of three distinct experiments and are expressed as percentages of the AFB1 produced by untreated *A. flavus* control cultures. Significant differences between concentrations of the same plant extract are indicated with *** for *p*-value < 0.001.

**Figure 6 molecules-27-08550-f006:**
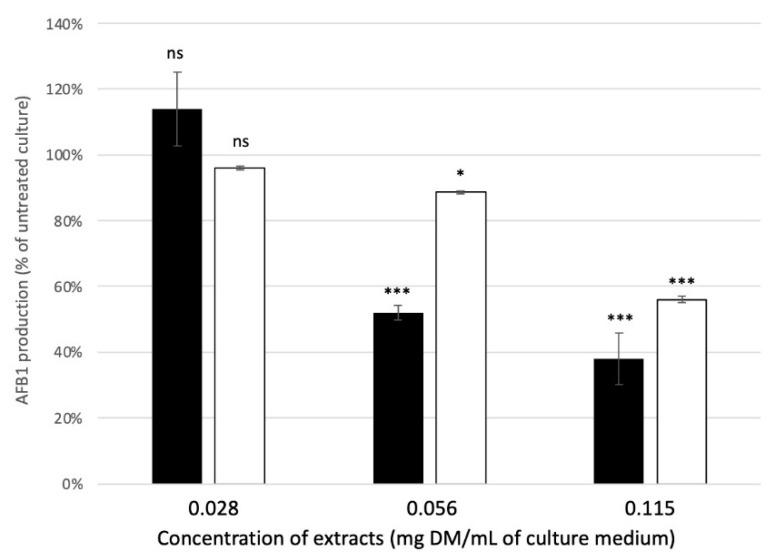
AFB1 production when exposed to low concentrations of mate (black bars) and green tea (white bars). Results correspond to the mean +/− SD of three distinct experiments and are expressed as percentages of the AFB1 produced by untreated *A. flavus* control cultures. Significant differences between concentrations of the same plant extract are indicated by * for *p*-value< 0.05, and by *** for *p*-value < 0.001. ns: not significant.

**Figure 7 molecules-27-08550-f007:**
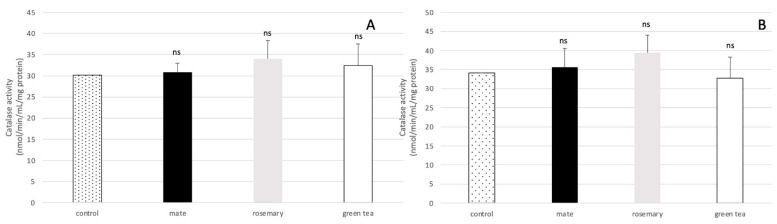
Impact of plant extracts on the catalase activity in *A. flavus*. (**A**) The fungal strain was exposed to 0.225 mg DM/mL of culture medium of extracts of mate (black bars), rosemary (grey bars), and green tea (white bars) or (**B**) to the IC80 of the three extracts that corresponded to concentrations of 0.16, 1.11, and 0.28 mg DM/mL of culture medium for mate, rosemary and green tea, respectively. Control (bars with black dots) corresponds to catalase activity in *A. flavus* culture with no extract. Results are given as the mean +/− SD of 6 replicates. ns: not significant.

**Figure 8 molecules-27-08550-f008:**
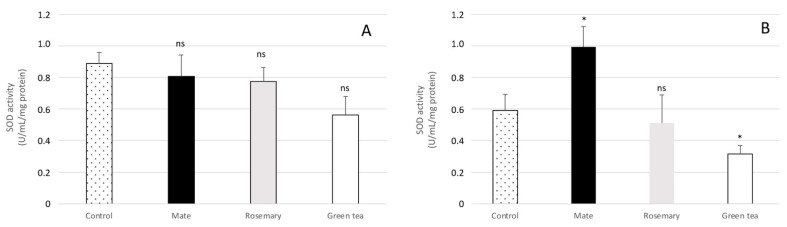
(**A**) Impact of plant extracts on the superoxide dismutase activity in *A. flavus*. The fungus was exposed to 0.225 mg DM/mL of culture medium of extracts of mate (black bars), rosemary (grey bars) and green tea (white bars) or (**B**) to IC80 of the three extracts corresponding to concentrations of 0.16, 1.11, 0.28 mg DM/mL of culture medium for mate, rosemary, and green tea, respectively. Control (bars with black dots) corresponds to superoxide dismutase activity in *A. flavus* culture with no extract. Results are given as the mean +/− SD of six replicates. Significant differences with untreated control are indicated by * for *p*-value <0.05. ns: not significant.

**Table 1 molecules-27-08550-t001:** Impact of plant extract concentration on growth of *A. flavus* as measured by colony diameter after 7 days of culture at 27 °C.

	Fungal Growth		
	(% of Untreated Control)		
	Mate	Rosemary	Green Tea
Concentration of extract			
(mg DM/mL ^a^)			
0	100%	100%	100%
0.0281	96% ***	nt ^b^	96% **
0.0563	97% *	nt	97% **
0.113	96% **	nt	96% **
0.225	98% ***	98% **	88% ***
0.45	96% **	96% **	91% **
0.9	99% **	97% *	95% *
1.8	95% *	94% ***	92% **
3.6	100% **	92% **	90% *
7.2	98% **	84% **	82% *

Shown are the means of six replicates; ^a^: mg DM/mL: mg of dry mater/milliliter of culture medium; ^b^: nt: not tested. Significant differences with untreated control are indicated by * for *p*-value< 0.05, ** for *p*-value < 0.01, and *** for *p*-value < 0.001.

**Table 2 molecules-27-08550-t002:** Pearson correlation coefficients between anti-aflatoxigenic potential and polyphenol content, global antioxidant activity and antiradical activity for mate, rosemary, and green tea extracts.

	Anti-Aflatoxigenic Activity		
	Mate	Rosemary	Green Tea
Polyphenol content	0.825	0.753	0.935
Antiradical activity	0.711	0.660	0.853
Antioxidant activity	0.5	0.668	0.660

## Data Availability

Data are available upon reasonable request to the corresponding author.

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
