# Peer review of "Inhibition of Aflatoxin B1 Synthesis in Aspergillus flavus by Mate (Ilex paraguariensis), Rosemary (Rosmarinus officinalis) and Green Tea (Camellia sinensis) Extracts: Relation with Extract Antioxidant Capacity and Fungal Oxidative Stress Response Modulation"

_molecules, 2022, doi:10.3390/molecules27238550_

Round 1
Reviewer 1 Report
Line 21 and 30: AFB1, SOD, etc., explain all abbreviations the first time they are used in abstract, text, tables, figures…
Lines 24-25: The expression units for the plant extracts (μg Dry Mater (DM)/mL) are not well understood. Usually, the dry matter or dry weight is a measurement of the mass of something when completely dried and expressed as mg/g dry matter or μg/g dry matter). It is confusing since in some parts of the manuscript concentration units are expressed as μg or mg per gram of extract and in other parts as μg or mg per mL of extract. Also, the wording μg Dry Mater (DM)/mL is not standard. It should be explained somewhere how to convert mg/mL to mg/g dry weight.
Line 58: microbiota or mycobiota is preferred over microflora
Lines 89 and 423: perhaps the word ‘demonstrate’ could be substituted for showed, reported or something similar
Figure 1: The text on the x-axis is cut off in some places.
Lines 113-115: In text mg GAE/mL but in Figures 1 and 2 mg GAE/g. How was the conversion from fresh weight to dry matter done? (the sequence of values 10.25, 2.5 and 5.29 do not match with 53.5, 24.3 and 102.1 mg GAE/g Dry Matter, respectively), was the order changed?
Line 120: homogenize the use of common names or scientific names (preferred) for plants (not mix i.e. Baphia nitida or dates) throughout the whole manuscript
Figure 2: why represent acetonitrile:water extracts if the mixture acetone:water worked best (according to line 98)? In fact, it seems that acetonitrile was not tested anyway (M&M section)
Figure 3: the units for the x-axis are missing. Revise all figures in manuscript as some of them have missing parts.
Line 154: Is the same 1.8 mg DM/mL than 1.8 mg/ g dry weight?
Table 1: None of the extracts showed a strong effect on radial growth of Aspergillus flavus at any of the concentrations tested. Is that the reason why MIC and MFCs values were not calculated? Why were higher concentrations not tested?
Lines 188-189: Please improve the wording of this sentence
Material and method: where did the plant material used, i.e. mate, rosemary and green tea leaves, come from? It is said that in the case of green tea, the leaves were used to obtain the extracts, but what plant parts were used for mate and rosemary?
The actual concentrations of active compounds are not clear as the expression units are ill defined (fresh basis and dry matter).
Usually, MIC and MFC values are calculated in vitro assays of the antifungal activity of plant extracts. Is IC50 (IC80) related to MIC and MFC values?
HPLC-FLD analysis: what was the retention time of AFB1?
Author Response
The authors would like to thank the Reviewer for its very relevant comments and careful reading of our manuscript. All of his remarks were taken into account and the text was improved accordingly

Reviewer 2 Report
In this paper, a higher correlation was found between anti-AFB1 activity and radical scavenging capacity of extracts, together with a modulation of SOD induced by mate and green tea in Aspergillus flavus. Rutin, a phenolic compound present in the three tested plants was shown to inhibit AFB1 synthesis and may be responsible for the anti-mycotoxin effect reported here.
The topic of this article is clear, rational, rich content and strong innovation. However, some minor issues still need to be improved. After minor revision, it can be accepted.
1) The author should check the whole manuscript for any grammatical errors and any other differences. Writing needs considerable improvement.
2) Line 108 and 148, change “p” to “P”. Please check this manuscript to avoid similar errors.
3) Please check the format of the text in the manuscript to make sure it is consistent, such as Line 110 and 128.
4) Line 152, change “on” to “in”. Please check this manuscript to avoid similar errors.
5) Line 256, change “A. flavus” to “A. flavus”, and italics should be used. Please check this manuscript to avoid similar errors.
6) Line 302, add “.” at the end of the sentence.
7) Line 340, change “Fifty” to “50”.
Author Response
The authors would like to thank the reviewer for his careful reading of our paper and comments. All remarks were taken into account and manuscript was improved accordingly.

Round 2
Reviewer 1 Report
I believe that the authors have substantially improved the paper in accordance with the reviewer's suggestions and have satisfactorily answered all the issues raised. In addition, the English revision seems to me very adequate. For all these reasons, I recommend the publication of the present revised version in the journal Molecules.